# How facemasks shape trust in social interactions

**Junning Peng**[1]*, **Andrea Isoni**[1], **Ashley Luckman**[2], **Hossam Zeitoun**[1], **Ivo Vlaev**[1], **Dawn Eubanks**[1], **Daniel Read**[1]

1 Warwick Business School, University of Warwick, Coventry, United Kingdom, 2 University of Exeter Business School, University of Exeter, Exeter, United Kingdom

* Junning.peng@warwick.ac.uk

## Abstract

Face coverings can potentially impact how trustworthy someone appears through two channels: by hiding important facial cues associated with trust; or by signalling the wearer's intentions or personal characteristics. The facemasks widely adopted during the COVID-19 pandemic both obscured the face and were associated with pro-social attitudes or intentions. The goal of this paper is to investigate how facemasks impact judgments about the trustworthiness of the wearer, and whether this would affect interactions with others. We report three experiments. Experiments 1 and 2 examined decisions in a two-player trust game when participants interacted with a single masked or unmasked counterpart. Experiment 3 explored whether participants were more likely to trust a masked or an unmasked person in a straight choice between them. In all experiments, masked faces were judged more trustworthy than unmasked ones. While in Experiments 1 and 2 this was not reflected in trust behaviour, in Experiment 3 over 70% of participants chose to trust the masked person, a decision predicted by the difference in perceived trustworthiness between the masked and unmasked counterparts. This suggests that in settings where facemasks or similar trust related cues are more salient, such as in joint evaluation, they can lead to enhanced trust.

## Introduction

During the COVID-19 pandemic, many public health policies were developed to reduce the impact of the virus. One widely adopted policy was a mandate on the wearing of protective facemasks in public spaces. Although such mandates are no longer in force, the use of facemasks by vulnerable people or those affected by respiratory diseases is still more common than in pre-pandemic times. Since facemask wearing was effective in reducing the spread of the virus (e.g., [1]), it will likely be adopted in response to future epidemics or pandemics (e.g., [2]). Moreover, face masks are likely to be used as public health responses to other air quality issues such as pollution, and by the public in response to seasonal concerns such as the common cold and flu (e.g., [3]).

**Data availability statement:** All relevant data are within the manuscript and its Supporting Information files.

**Funding:** This work was supported by the Economic and Social Research Council via the Network for Integrated Behavioural Science [award number: ES/P008976/1]. The funders had no role in study design, data collection and analysis, decision to publish, or preparation of the manuscript.

**Competing interests:** The authors have declared that no competing interests exist.

Unlike many health-related actions, such as vaccination or taking pills, a mask is highly visible and its effects may therefore extend well beyond individual and community health [4–6]. One suggestion is that mask wearing changes how people interpret themselves, with the mere act of putting on a protective mask making people feel more ethical (e.g., [7,8]).

In this paper, we focus on a specific relationship between mask wearing and ethicality: the influence of mask wearing on how people assess whether others are trustworthy, and the consequent willingness to actually place trust in others. We do this by conducting experiments based on variants on the trust game [9,10], widely used to assess the role of trust in decision making.

There are two channels through which facemask wearing might influence trust. The first relates to the role of facial cues in interpersonal interactions, and the second concerns the cultural associations between mask wearing and trustworthiness. With respect to the first channel, facial cues are amongst the central factors used to assess trustworthiness (e.g., [11–16]). However, judgments of trustworthiness may be less reliable, or at least held with less confidence, when masks obscure important facial cues. While there is no direct evidence on this, there is evidence that judgments of emotion, likely to be related to trustworthiness, are more difficult and less reliable when facial cues are occluded (e.g., [17]). Relatedly, Grahlow et al. [18] found that people were less confident in their judgments of emotions when evaluating masked versus unmasked people. The same may apply to trustworthiness judgments. Indeed, even if the judgments themselves are not impaired objectively, because trust depends on openness, it may be harder to achieve when the information it rests on is unavailable.

The second channel relates to the cultural associations between mask wearing and individual characteristics. In some places, face coverings are strongly associated with criminality [19]. In others, they are associated with religious status [20], and in yet others they demonstrate social solidarity [2,21]. Many of these associations are themselves related to trustworthiness, although the direction of that association varies.

Several recent studies have linked the wearing of facemasks to judgments of trustworthiness. In line with the literature on face coverings and emotion identification (e.g., [18]), Marini et al. [22] found that people were less good at distinguishing between a set of standardised trustworthy and untrustworthy faces when those faces wore masks. This suggests a role for the first of the two channels we identified, with masks making it more difficult to assess trustworthiness. On the other hand, other studies have found that masked faces were typically judged as more trustworthy than unmasked ones [23–25]. This could be related to the general perception that those who wear masks care for others and are attempting to reduce the risk of inadvertently spreading disease [26–28].

Because mask wearing and trust appear related, it is important to investigate whether this relationship carries over into interactions that rely on trust. We addressed this with three experiments based on the trust game, a widely used experimental game that simulates many everyday interactions in which one party must put faith in another to achieve a higher reward [9,10,29]. In the trust game, one party

(the *trustor*) can make themselves vulnerable to another party (the *trustee*) to open-up opportunities for mutual gain. The game is a stylised model of situations such as asynchronous market transactions, or investing in a joint enterprise based on a handshake, where one party must act on the assumption that the other will reciprocate. In Experiments 1 and 2, we investigate whether the behaviour of trustors is influenced by whether the trustee is wearing a surgical facemask. We also test whether facemask wearing by trustors influences the trustees' behaviour. In Experiment 3, we study whether masked counterparts are more likely to be trusted than unmasked ones.

## Materials and methods

### Overview of the experiments

We conducted three online experiments with UK residents to investigate the influence of facemasks on trust and per-ceived trustworthiness. Because the UK lifted lockdown restrictions before many other countries, while retaining mask mandates and recommendations, at the time of our experiments there were many opportunities for individuals to interact with each other while one or both were wearing facemasks. As already mentioned, the experiments were based on the trust game. Experiments 1 and 2 deployed what Kugler et al. [10] call the complex trust game, and Experiment 3 a variant of what they call the basic trust game. The decisions in the experiments were hypothetical, and participants were informed of this.

Experiment 1 was conducted on 9th June 2021, a few weeks the UK government announced the reopening of most indoor venues (i.e., pubs, restaurants, cinemas) and allowed up to 30 people to mix in outdoor settings. Experiment 2 was conducted on 14th October 2021, when the UK was again experiencing a large volume of daily confirmed infection cases and facemasks were still mandated on public transport and in most public indoor places. Experiment 3 was conducted on 2nd July 2022, about 6 months after the mandatory facemask policy was lifted, and when the great majority of adults and teenagers had received at least two vaccinations against COVID-19 [30]. To control for changes in public attitudes across these time points, we assessed participants' attitudes toward facemasks in each experiment.

### Ethics statement

All experiments were approved by the Humanities and Social Sciences Research Ethics Committee of the University of Warwick. Participants provided informed written consent electronically through the online recruiting platform, Prolific, before beginning each experiment. Consent was documented digitally via the Prolific platform. No participants under the age of 18 were invited to this experiment. We ensured that the sample for each experiment was unique by inviting only participants who had not participated in our previous experiments. Preregistrations for all experiments, analysis plans, complete data and other related materials can be found at https://osf.io/yrd7j/.

All experiments investigated the influence of facemask wearing on participants' decisions in variants of the trust game. Since Experiments 1 and 2 had very similar designs, they are presented together. Experiment 3 was set up differently and is presented separately.

## Experiments 1 and 2

### Participants

Participants were recruited online through Prolific (https://www.prolific.co/). We pre-registered sample sizes based on our budget and specified 100 participants per experimental condition. Post-hoc power analyses indicate this gave us a power of 0.9 to detect primary effects, i.e., the effects of masks on trustor behaviour, of at least 0.01 in Experiment 1 and 0.03 in Experiment 2. We recruited 804 participants (402 male, 402 female) with a mean age of 33.6 years (SD = 12.1, range 18–71) in Experiment 1, and 402 participants (201 male, 201 female) with a mean age of 31.9 years (SD = 10.8, range 18–78) in Experiment 2. All participants received £0.70 for an average of approximately seven and a half minutes. In

Experiment 1, after applying our exclusion criteria (detailed in the following section), we retained 718 out of the initial 804 participants. Similarly, in Experiment 2, using the same criteria, we kept 361 out of the 402 participants.

## Experimental design and procedure

In the main task, participants were introduced to the trust game and were informed they should decide as if they were playing with the counterpart whose picture was displayed on their screen. The game was played only once. The characteristics of the counterpart (i.e., masked and unmasked) and the gender of the participant were our main independent variables. As detailed in the next section, our manipulation of the counterpart consisted in showing pictures of a hypothetical co-player. For this reason, we could not provide truthful real-time feedback on the outcome of the game, nor could we link the participants' decisions in the game to monetary incentives. Thielmann et al. [31] show that behaviour in hypothetical trust games is indistinguishable from behaviour in real-incentive versions when all other design features are kept constant. For the same reason, the game was played only once, as is common in the literature [9,32,33].

Experiment 1 used a two (masked vs unmasked counterpart) by two (male vs female counterpart) by two (male vs female participant) between-subjects design. Experiment 2 used a two (masked vs unmasked counterpart) by two (male vs female participant) between-subjects design.

In our version of the trust game, the trustor is endowed with £10. The trustor decides how much of this £10 to send to the trustee ($s$, which can take any integer value from 0 to 10). The amount sent is tripled, and the trustee then decides how much of the tripled amount, if any, to return to the trustor ($r$, which can take any integer value between 0 and $3s$). Consequently, the trustor's payoff equals ($10 - s + r$), and the trustee's equals ($3s - r$). The amount sent by the trustor is generally interpreted as a measure of how much the trustor trusts the trustee, and the amount returned by the trustee as a measure of their trustworthiness [9,32,34]. To ensure that the trustor would not worry that the trustee would keep everything because they had no other earnings, we also indicated the trustee was initially endowed with £10 (see Berg et al. [9] and other implementations of the trust game).

The experimental instructions described the trust game as having two "stages", one for the trustor's decision ("Sender" in the instructions) and one for the trustee's decision ("Responder"). To aid understanding, after each stage was explained, participants answered four multiple-choice questions about the instructions and received immediate feedback that rectified any misunderstanding.

Each participant was randomly assigned to either a masked or unmasked counterpart and then took part as both trustor and trustee, with role order counterbalanced. Evidence suggests that people who know in advance they will be playing both roles in the trust game typically become more selfish [35], so we minimised this order effect by not announcing the presence of a second decision until it was encountered.

The trustor's decision consisted of selecting an integer amount from the interval [£0, £10] that would be sent to the trustee. For the trustee decision, we used the strategy method (e.g., [36]), asking participants to indicate how much they would return conditional on each of the eleven amounts they could have received from the trustor.

After completing the trust game, participants answered four comprehension check questions. As preregistered, anyone who answered any of the first two questions incorrectly was excluded from the analysis. The first two questions were simple and served as "attention checks". For example, "Imagine the Sender sends £3. How much does this become before it is received by the Responder?". The second two questions were more advanced and were designed to reinforce the instructions – "Based on the information from the previous questions, and recalling that the Sender received £10 at the beginning of the interaction, how much do they earn in total?" All four questions with answer options are reproduced in the Supplementary Materials S3 File.

Participants next rated the attractiveness and the trustworthiness of the counterpart depicted in the picture on a scale from 0 to 100. They then reported their vaccination status and responded on a 0–10 scale regarding their beliefs in the efficacy of facemasks for COVID-19 prevention and their perceptions of facemask usage within the local community. They

also expressed their views on whether facemasks serve to protect the wearer or others, and the frequency with which they wore facemasks themselves. These were both assessed on a 5-point Likert scale. Finally, participants stated how they voted in the most recent UK general election.

Experiment 2 followed the same procedures as Experiment 1, with two changes. First, to increase the salience of the counterpart's face, we required participants to view their picture for at least 10 seconds at the start of the experiment. Second, since the gender of the hypothetical counterpart had no influence on the behaviour of either trustors or trustees in Experiment 1, we restricted Experiment 2 to female counterparts only. That the gender of the trustee has no effect on trust and trustworthiness is consistent with a recent meta-analysis [37].

**Face stimuli.** The images for the experimental counterparts were obtained from the Chicago Face Database (CFD) (https://chicagofaces.org/default/), which consists of 597 high-resolution, standardised portrait photographs of men and women demonstrating neutral and other expressions. The CFD also provides numerical measurements of 32 facial attributes (e.g., nose length) and 29 subjective judgments (e.g., sadness) including trustworthiness ratings. To control for the possible influence of ethnic background and facial expression on trustworthiness, we selected white faces (the largest ethnic group in the UK) with neutral expressions.

Because we did not want visible facial features and expressions to confound the influence of facemasks, we selected four male and four female faces using a procedure that minimised differences in age, attractiveness and trustworthiness as recorded in the CFD. These faces were chosen to look like ordinary people participants would routinely see in their daily lives (their CFD unusualness ratings ranged from 2.1 to 3.1 on a 1–7 scale, 1 = Not at all; 7 = Extremely). Moreover, we found 73 studies mentioned on Google Scholar that used items from the CFD on Prolific. Of these, 32 were conducted on UK samples. These generally involved small samples of participants (100 or so) and a small number of faces. Given that Prolific has around 80,000 unique and active UK participants, we suggest the likelihood that our participants would recognise any of these faces is negligible.

To obtain the masked faces for our mask manipulation, a standard surgical mask was superimposed on each selected picture using photo editing software. This is a commonly used method to maintain strict stimulus control (e.g., [38–40]). Some examples of the resulting masked and unmasked faces are shown in Fig 1. Full details on how the faces were selected and the corresponding masked and unmasked images can be found in the Supplementary Materials S1 File.

## Results

**Trustors' decisions.** Fig 2 shows the average amount sent by trustors under different conditions in Experiments 1 and 2. Separate bars indicate whether the trustor was paired with a masked (dark) or unmasked (light) counterpart. For Experiment 1, counterpart genders are presented in different rows. The figure shows that the amount sent by the trustors was not influenced by the presence of the mask, and this effect was consistent across different treatment conditions in both experiments.

In line with our preregistration, we performed linear regressions to test whether the facemask influenced how much trustors sent (Table 1). As might be inferred from Fig 2, the facemask did not have a significant effect in either experiment (Exp 1: $t(710) = 1.02$, 95% CI [–0.37, 1.19], $d = –0.01$, $p = .306$, Exp 2: $t(356) = 0.09$, 95% CI [–0.89, 0.98], $d = –0.05$, $p = .926$). This suggests that wearing a facemask had no effect on trust.

Participant gender had a significant effect in Experiment 1, with men sending an average of £1.65 more than women (Exp 1: $t(710) = 4.15$, 95% CI [£0.87, £2.43], $d = 0.44$, $p < .001$), a difference that was still significant after splitting by counterpart's gender (for male counterparts: $t(352) = 2.98$, 95% CI [£0.46, £2.25], $d = 0.43$, $p = .003$, for female counterparts: $t(356) = 4.08$, 95% CI [£0.97, £2.76], $d = 0.44$, $p < .001$). Although the pattern was similar in Experiment 2, the effect size was smaller and not statistically significant (Exp 2: $t(356) = 1.31$, 95% CI [£–0.31, £1.55], $d = 0.20$, $p = .190$). In both experiments, the amount sent by trustors was lower if they first took the role of trustee (Exp 1: $t(710) = 4.61$, 95% CI [0.61, 1.52], $d = 0.33$, $p < .001$; Exp 2: $t(356) = 4.57$, 95% CI [0.88, 2.20], $d = 0.47$, $p < .001$). Separate analyses conducted on

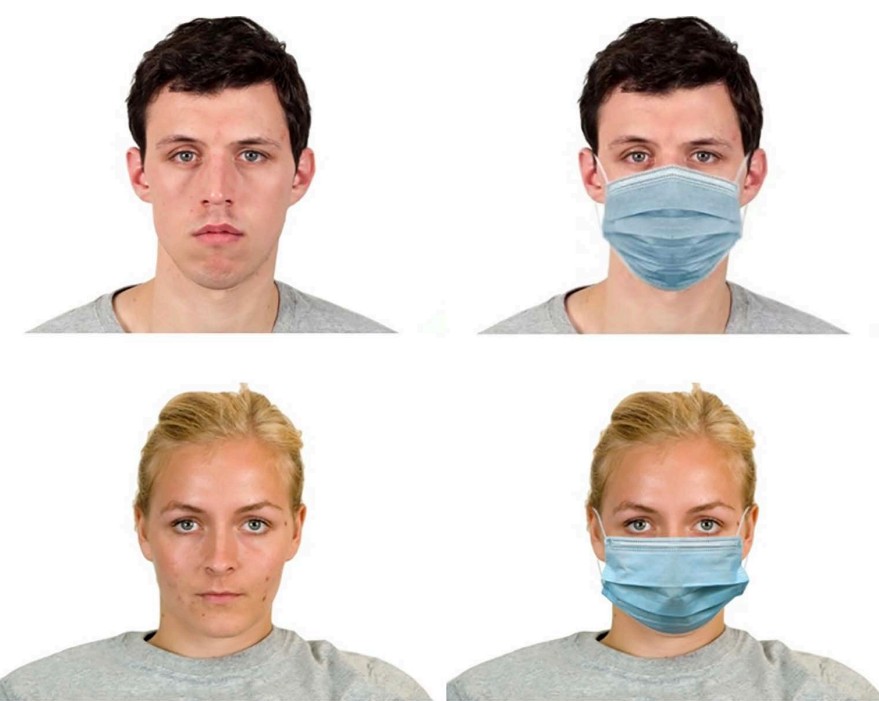

**Fig 1. Examples of male and female counterparts, both masked and unmasked.**

the first and second decisions similarly show no effect of the mask manipulation (see Supplementary Materials S8 File). One possibility is that experiencing the temptation to be "greedy" as a trustee made participants recognise that others might also be tempted. Burks et al. [36] similarly found that trustors who knew they were going to play both roles sent less money than those who did not.

**Trustees' decisions.** Fig 3 shows the average proportion of the money received that the trustees returned to the trustor as a function of the amount received. There was an upward trend in both studies, suggesting that trustees returned a higher proportion as the amount sent by the trustor increased. In both experiments, the proportion returned was not influenced by whether the trustor was masked or unmasked.

As preregistered, we analysed the trustees' decisions using multi-level regressions. This was done because each participant provided several responses, one for each amount sent by the trustor, and these responses were not independent. Multi-level regressions account for this lack of independence. To make all the responses comparable, we use the proportion returned as our dependent variable (for example, if a trustor sent £5, the trustee received £15, and returned £3, we coded this as a return of 20%). This is a standard approach when analysing the trust game (e.g., [41]). We included fixed effects for participant gender, counterpart gender, facemask manipulation, the amount sent by trustors, role order (trustee or trustor first), and all two-way interactions, as well as random effects for participants.

As with the trustors, the facemask manipulation showed no significant influence on trustees' decisions (Exp 1: $t(709)$ = –0.82, 95% CI [–0.06, 0.02], $p$ = .413; Exp 2: $t(355)$ = –0.34, 95% CI [−0.06, 0.04], $p$ = .737). However, the proportion returned was positively correlated with the amount sent by the trustor, with each extra £1 sent increasing the proportion returned by 1.7% in Experiment 1 ($t(709)$ = 7.20, 95% CI [0.93%, 1.89%], $p$ < .001) and 1.1% in Experiment 2 ($t(355)$ = 3.37, 95% CI [0.45%, 1.71%], $p$ = .001). This is consistent with the idea that "trust begets trust", as suggested by Cohen and Isaac [42].

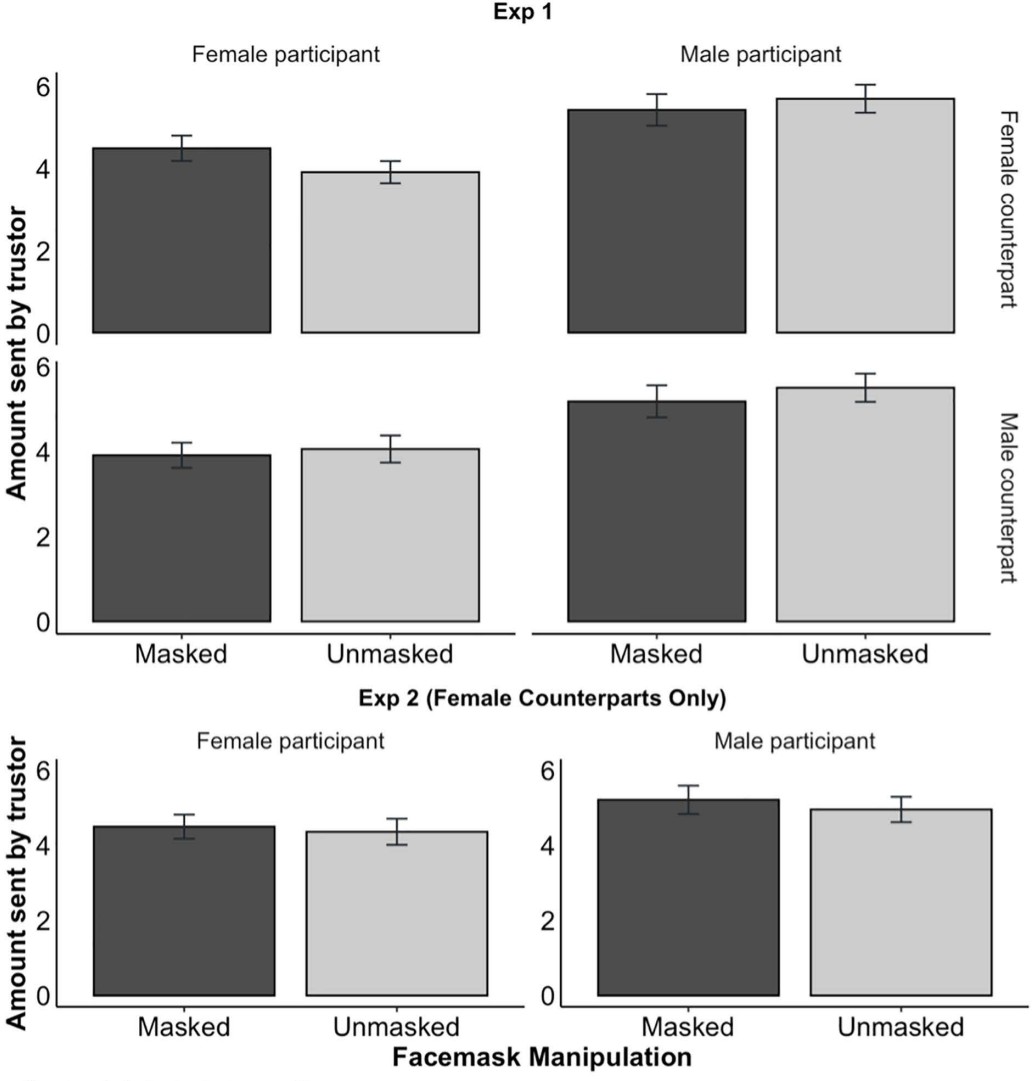

Fig 2. Average amount sent by trustors in Experiments 1 (top) and 2 (bottom), as a function of whether their counterpart was Masked (dark) or Unmasked (light), and whether the participant was Male (right column) or Female (left column). Error bars indicate standard errors of the mean.

The order in which the roles were played had a significant effect on the proportion returned by trustees in Experiment 1 ($t(709) = -2.00$, 95% CI [-5.15%, -0.33%], $p = .046$), but not in Experiment 2 ($t(355) = -0.24$, 95% CI [-3.93%, 3.10%], $p = .806$). In Experiment 1, participants who played as trustee first returned on average 2.5% more than those who played as trustor first. This effect – akin to Johnsen and Kvaløy's [43] finding of less reciprocation of trust if it was announced to participants that they would play two repetitions of the game than if it was not – did not interact with our mask manipulation (see Supplementary Material S2 File). Full regression results are shown in Table 2. As a robustness check, S9.1 Table reports separate regression estimates for each amount that could be sent by the trustor. When controlling for participant gender, counterpart gender, and order of play, the proportion returned by the trustee is consistently unaffected by the mask manipulation.

**Table 1. Linear regression predicting the amount sent by trustors as a function of participant's gender (Male participant), counterpart's mask status (Masked picture), counterpart's gender (Male picture), and the order in which the participant completed the tasks (Played trustor first).**

| Predictors | Experiment 1 | | | Experiment 2 | | |
|---|---|---|---|---|---|---|
| | Estimates | CI | p | Estimates | CI | p |
| Masked picture | 0.41 (0.398) | [−0.37, 1.19] | 0.306 | 0.04 (0.382) | [−0.89, 0.98] | 0.926 |
| Male participant | 1.65 (0.398) | [0.87, 2.43] | <0.001 | 0.62 (0.474) | [−0.31, 1.55] | 0.190 |
| Male picture | 0.07 (0.394) | [−0.70, 0.84] | 0.856 | – | – | – |
| Played trustor first | 1.07 (0.231) | [0.61, 1.52] | <0.001 | 1.54 (0.337) | [0.88, 2.20] | <0.001 |
| Male picture * Masked picture | 0.40 (0.462) | [−1.31, 0.51] | 0.385 | – | – | – |
| Male participant * Masked picture | 0.46 (0.462) | [−1.37, 0.45] | 0.321 | −0.19 (0.674) | [−1.14, 1.51] | 0.782 |
| Male participant * Male picture | −0.09 (0.462) | [−1.00, 0.81] | 0.839 | – | – | – |
| Observations | 718 | | | 361 | | |
| $R^2$ | 0.077 | | | 0.065 | | |
| Adj. $R^2$ | 0.068 | | | 0.055 | | |
| AIC | 3667.3 | | | 1869.2 | | |

**Exploratory analysis of the effect of mask-wearing on trustworthiness ratings.** We looked at participants' trustworthiness ratings of the face stimuli. These were expressed on a 0–100 slider where the numeric value was not visible to participants. In both experiments, average trustworthiness ratings were higher for masked counterparts: 8.32 and 9.95 points higher in Experiments 1 and 2 respectively (see Fig 4). This was confirmed with linear regression (Exp 1: $t(703) = 3.03$, 95% CI [2.93 points, 13.71 points], $d = 0.25$, $p = .003$; Exp 2: $t(346) = 2.76$, 95% CI [2.85 points, 17.05 points], $d = 0.29$, $p = .006$). In Experiment 1, in which we varied the gender of the counterpart, female faces were rated on average almost 8 points higher on trustworthiness than male faces ($t(703) = −2.78$, 95% CI [−13.02 points, −2.24 points], $d = 0.19$, $p = .006$). For detailed regression tables including this and other exploratory analyses see the Supplementary Materials S2 File.

**Additional exploratory analyses of trustor/trustee decisions.** In both experiments, higher trustworthiness ratings were associated with higher amounts sent by the trustor (Exp 1: $t(703) = 4.84$, 95% CI [£0.03, £0.06], p < .001; Exp 2: $t(349) = 2.37$, 95% CI [£0.00, £0.05], $p = .019$), with a one point increase in trustworthiness rating associated with an increase in the average amount sent of £0.04 in Experiment 1, and £0.03 in Experiment 2. For example, an increase in perceived trustworthiness from 30 to 70 points would increase the amount sent by £1.60 in Experiment 1 and £1.20 in Experiment 2. In both experiments, trustworthiness ratings were also positively related to the proportion returned by trustees (Exp 1: $t(706) = 5.72$, 95% CI [0.001, 0.003], $p < .001$; Exp 2: $t(351) = 4.27$, 95% CI [0.001, 0.002], $p < .001$).

We speculated that those who reported that facemasks primarily protect others might send more in the trust game because these beliefs may be related to altruistic traits. They may feel a greater sense of responsibility towards the wellbeing of others, and this could lead them to act more generously. In both experiments, participants reported who they believed masks protected on a 5-point scale (*only themselves, mostly themselves, equally themselves and others, mostly others, only others*). In Experiment 1, participants who reported that facemasks protect only or mostly the wearer sent £0.53 less ($t(703) = −2.03$, 95% CI [£−1.04, £−0.02], $d = 0.14$, $p = .043$) than those who said masks equally protect

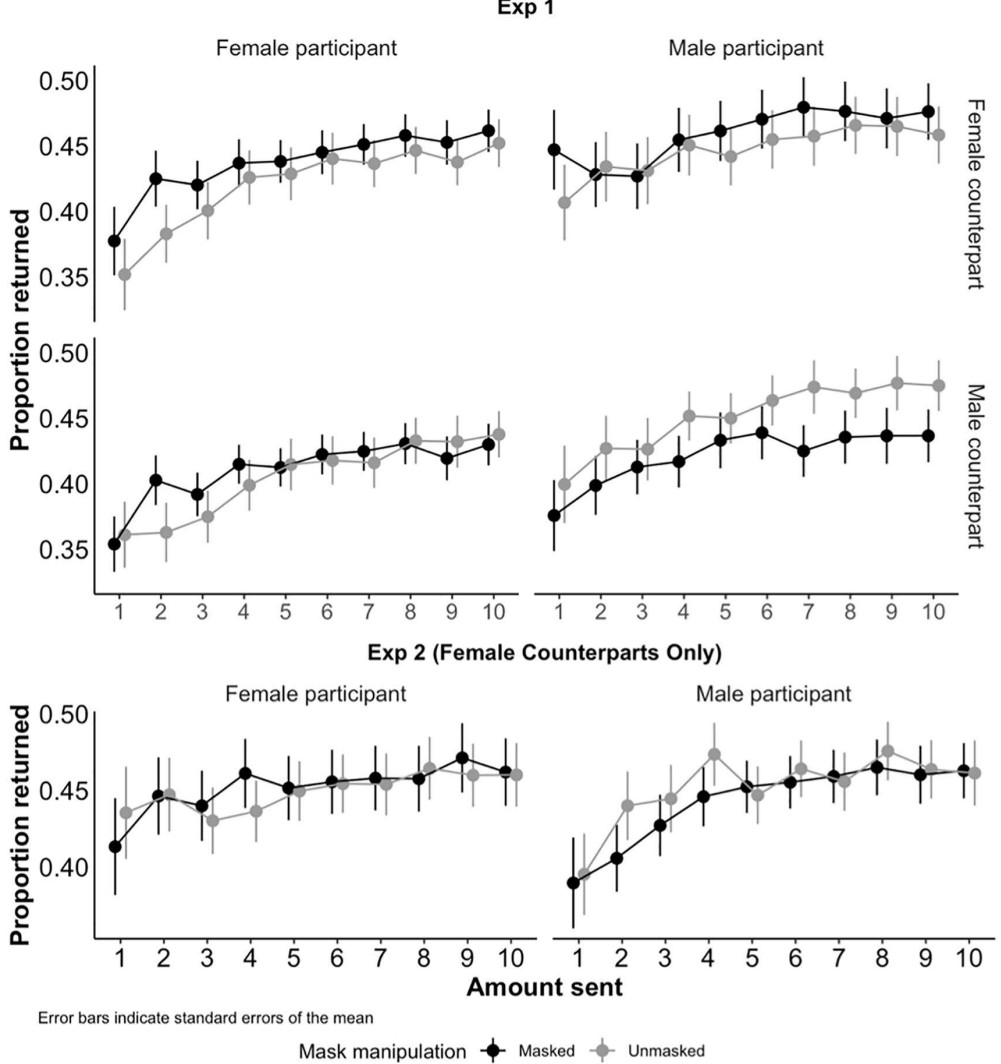

**Fig 3. Average proportion returned by trustees in Experiments 1 (top) and 2 (bottom) as a function of whether the counterpart was Masked (dark lines) or Unmasked (light lines), and whether the participant was Male (right column) or Female (left column).** Error bars indicate standard errors of the mean.

the wearer and others, and participants who reported that facemasks protect only or mostly others sent £0.60 more than those who said masks equally protect the wearer and others, but this effect was not significant ($t(703) = 1.85$, 95% CI [£–0.04, £1.23], $d=-0.27$, $p=.065$). In Experiment 2, there were no significant differences in any of these comparisons. One possible interpretation is that the pool of people reporting different attitudes toward masks had changed by the time of Experiment 2, and both findings reflect this historical shift.

Additionally, in Experiment 2 participants indicated whether they had changed their mask-wearing behaviour after the lifting of national restrictions on 19th July 2021. Irrespective of whether the counterpart was masked, Trustors who had not reduced their facemask usage sent an average of £0.68 more than those who had ($t(349) = 2.02$, 95% CI [£0.02, £1.35], $d=0.19$, $p=.044$).

**Table 2. Multi-level regression predicting the proportion returned by trustees as a function of participant's gender (Male participant), counterpart's mask status (Masked picture), counterpart's gender (Male picture), the order in which the participant completed the tasks (Played trustor first), and the amount sent by trustors.**

| Predictors | Experiment 1 | | | Experiment 2 | | |
|---|---|---|---|---|---|---|
| | Estimates | CI | p | Estimates | CI | p |
| Masked picture | −0.02 (0.020) | [−0.06, 0.02] | 0.413 | −0.01 (0.025) | [−0.06, 0.04] | 0.737 |
| Male participant | −0.01 (0.022) | [−0.05, 0.04] | 0.758 | −0.02 (0.025) | [−0.07, 0.03] | 0.457 |
| Male picture | −0.03 (0.021) | [−0.07, 0.01] | 0.168 | – | – | – |
| Played trustor first | −0.03 (0.013) | [−0.05, 0.00] | 0.046 | −0.01 (0.018) | [−0.04, 0.03] | 0.806 |
| Amount sent | 0.02 (0.002) | [0.01, 0.02] | <0.001 | 0.01 (0.003) | [0.00, 0.02] | 0.001 |
| Male picture * Masked picture | 0.02 (0.025) | [−0.03, 0.07] | 0.499 | – | – | – |
| Male participant * Masked picture | 0.02 (0.025) | [−0.03, 0.06] | 0.554 | 0.02 (0.035) | [−0.05, 0.08] | 0.670 |
| Male participant * Male picture | 0.02 (0.025) | [−0.03, 0.07] | 0.393 | – | – | – |
| Observations | 7180 | | | 3610 | | |
| Marginal $R^2$ | 0.074 | | | 0.029 | | |
| Conditional $R^2$ | 0.769 | | | 0.719 | | |
| AIC | −10538.8 | | | −4554.5 | | |

## Discussion

In Experiments 1 and 2, facemasks did not significantly affect the decisions of trustors (those sending money) or trustees (those returning money). This is despite participants rating masked faces as more trustworthy than unmasked faces, and both sending and returning more money to faces they rated as more trustworthy. There were gender differences in trustor behaviour, with males sending more than females. Participants who played the trustor role first sent more but returned less in Experiment 1, showing a shift towards less generous choices with game experience. This aligns with Kugler et al. [10], who found that consequential thinking reduces how much is sent, suggesting that playing the trustee role first engages such thinking, leading to more cautious decisions.

Given that trustworthiness ratings were higher for masked than unmasked counterparts, and that there was a positive relationship between those ratings and both the amount sent by trustors and the proportions returned by trustees, the lack of a main effect of the mask manipulation may seem surprising. Considering the explicit request, in Experiment 2, that participants focused on the picture of the counterpart for at least 10 seconds, we are confident that they were aware of the mask when this was present. We speculate that, because of the between-subject nature of our design (each participant only saw one face, either masked or unmasked) and the relative complexity of the trust-game scenario (the sequence of moves, the multiplication of the amount sent, the list of decisions to make as second mover), participants may not have perceived the presence of a mask as a sufficiently strong and relevant cue.

In Experiment 3, drawing on research into joint versus separate evaluation (e.g., [44]), we altered our design to make the presence or absence of a mask more salient and directly relevant to the participant's main decision, and simplified their decision task by focusing on the first mover's decision, which most directly hinges on perceived trustworthiness.

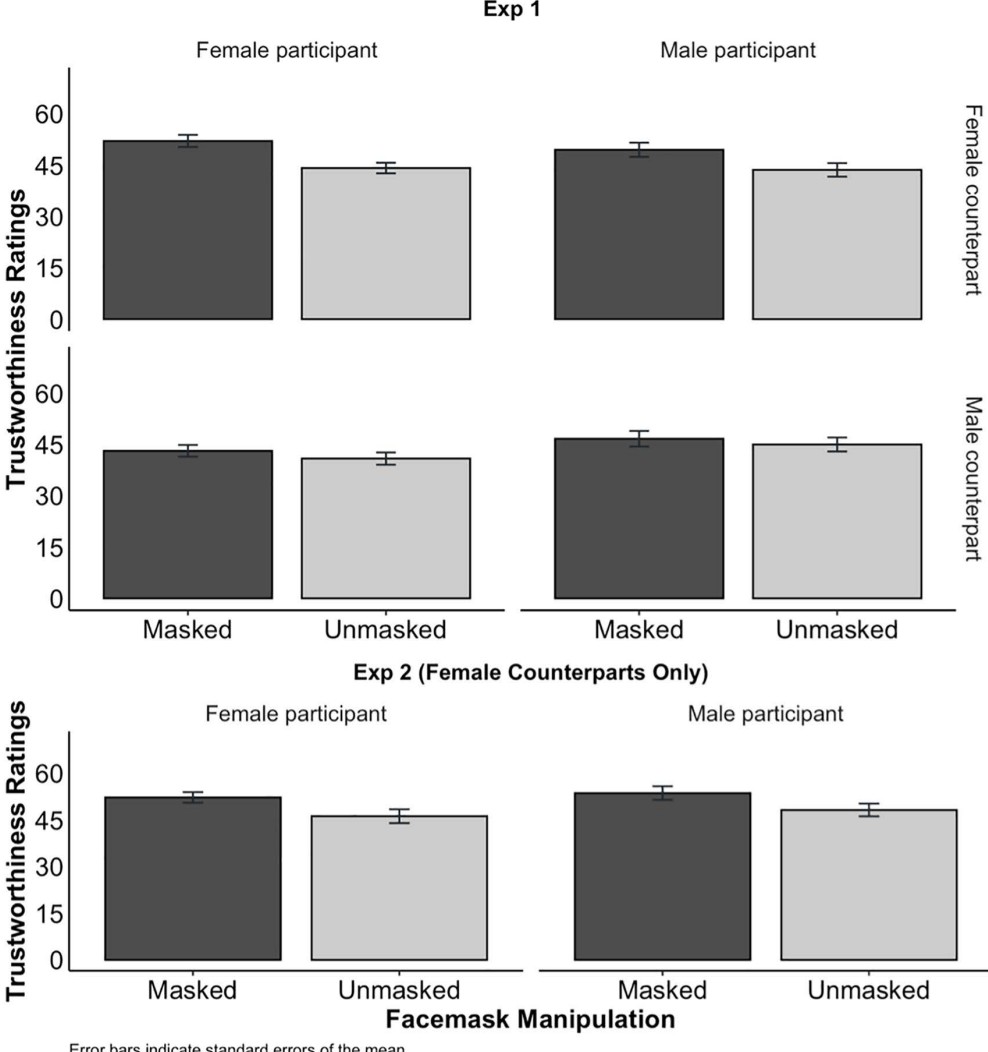

**Fig 4. Average trustworthiness ratings in Experiments 1 (top) and 2 (bottom), as a function of whether the counterpart was Masked (dark) or Unmasked (light), and whether the participant was Male (right column) or Female (left column).** Errors bars indicate standard errors of the mean.

## Experiment 3

For the difference in perceived trustworthiness between masked and unmasked counterparts in Experiments 1 and 2 to exert a systematic effect on the decisions made in the trust game, participants would have had to spontaneously think about a counterfactual counterpart that differed from the one they were facing in terms of mask wearing. In a relatively complex environment like the trust game, this may not always occur. In Experiment 3, we made the facemask more salient by presenting trustors with a choice that amounted to placing more trust either in a masked or an unmasked counterpart. We presented this choice in a simpler decision scenario.

The logic of this manipulation is based on the distinction between *joint* and *separate* evaluation, originally proposed by Hsee ( [44]; see also Bazerman et al. [45]). If someone allocates money to somebody else in the expectation of receiving some money back, they may want to allocate more if that person is more trustworthy. But in a separate evaluation task, when we have one person in front of us and we need to decide how much to give that person, we have no benchmark

concerning what "more" is, and indeed cannot compare that person to a less trustworthy other to decide that they should receive more than that other. More broadly, there may be situations in which trust decisions are inherently comparative – think about someone choosing between two traders in adjacent market stalls, one of whom may be wearing a mask. The mask may be saying something about the trustworthiness of the trader, which obviously matters for the perspective transaction. In Experiment 3, the participant's decision was framed to encourage a joint evaluation. Participants saw two counterparts, one masked and one unmasked, and decided who they wanted to allocate more to, knowing that each counterpart would have the opportunity to return some of the money received. We predicted the greater trustworthiness of the masked person demonstrated in Experiments 1 and 2 would be reflected in the choice between the masked and unmasked counterparts.

A further distinction between Experiment 3 and the earlier ones is in the trusting decision by the first mover. To simplify the task, we did not multiply the amount sent but left it unchanged, so that trustworthiness concerned how much of this money would be returned. The difference between participants is that one would have received £20, and the other £10. Because the trustor could not keep any money for themselves, their decision concerning who would receive the £20 was a decision about who could be trusted to send back more of the money. In this case, either the masked or unmasked person.

## Participants

We tested 199 participants (100 male, 99 female) with a mean age of 41.8 (SD = 14.2, range 18–79) in July 2022. The pre-registered sample size of 200 was based on budgetary constraints, but ex-post simulations show that it provides power greater than 0.9 for detecting a 65% preference for choosing the masked counterpart. Participants were paid £0.70 for their participation. Following the preregistered criteria, 28 participants were excluded for incorrectly answering at least one key comprehension question, leaving 171 for our analyses.

## Experimental design and procedure

A two (male vs female participant) by two (allocating £10 vs £20) between-subjects design was developed as an online experiment, called the money allocation task, which used elements of the trust game from Experiments 1 and 2. The participant, whom we continue to call the trustor, was to distribute an amount of money unevenly between two trustees, knowing that each trustee could return some of the money back to the trustor. We sought to discover if the trustor would give a higher amount to the masked or to the unmasked trustee.

The amount to be divided was £30, with £20 going to one trustee and £10 to the other. The two trustees could return some, none or all the money allocated to them. To reduce the complexity of the scenario, in contrast with the trust game of Experiments 1 and 2, the money allocated by the trustor was not multiplied. By requiring participants to make an unequal binary choice, the task allows us to identify who was trusted more by the trustee (the one given £20) and who less (the one given £10). Like in Experiments 1 and 2, to maintain control over the face stimuli, the decisions in this experiment were also hypothetical.

The same neutral-expression female pictures from Experiment 2 were used. We counterbalanced which person was masked and whether the masked face appeared on the left or the right of the screen. To control for the possibility that participants used the masked (or unmasked) face to break the tie between the two trustees, we implemented two choice frames. In the £10 frame, participants selected who would receive the lower amount of £10 (with £20 automatically going to the other person), and in the £20 frame, they selected who would receive the large amount of £20. Varying the choice frame also discouraged participants from assuming an expected response, ensuring that their allocation decisions were based on their judgment rather than any assumptions they might have had about the experiment.

Following this allocation task, all participants rated the attractiveness and trustworthiness of both faces individually (in counterbalanced order). Next, they stated whether they thought facemasks were worn to protect the mask wearer or other people and described their current mask-wearing habits.

## Results

**The money allocation task.** Fig 5 shows how often the masked counterpart was allocated £20 as a function of the participant's gender and how the allocation task was framed. Across conditions, between 69% and 74% of participants allocated £20 to the masked counterpart.

We ask whether the proportion allocating £20 to the masked counterpart exceeds 50%, and whether this depends on the participant's gender and the task frame. As pre-registered, we do this using logistic regressions (see Table 3) where the dependent variable is whether the masked counterpart was allocated £20. Because our benchmark is 50%, the main variable of interest is the intercept. After contrast coding for participant gender and frame (i.e., whether they were asked to allocate either £10 or £20) the logistic regression re-centres the categorical variables so that the intercept can be interpreted as the log-odds of giving £20 to the masked (vs. unmasked) person, averaging across participant gender and frame. In log-odds terms, intercept would be 0 if trustees were equally likely to give £20 to the masked trustee, and

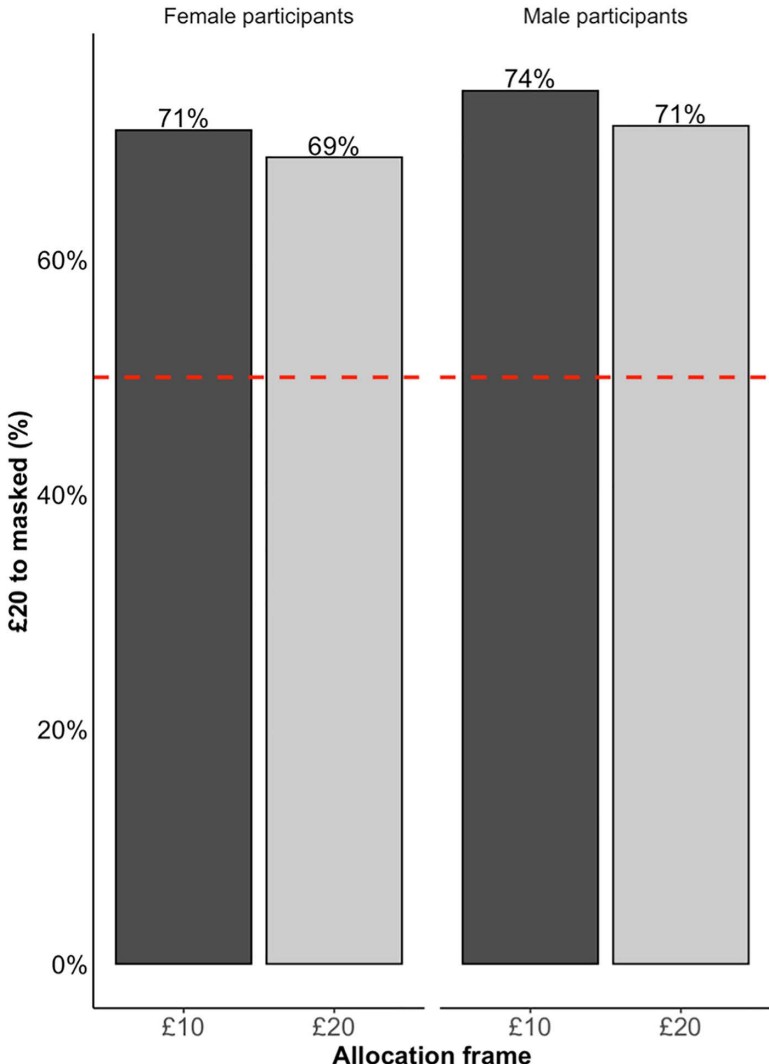

**Fig 5. The percentage of participants who allocated £20 to the masked counterpart as a function of whether the participant was female (left) or male (right), and whether they were allocating £10 (£10 frame, dark bars) or £20 (£20 frame, light bars).**

greater than 0 if they were more likely to give it to them (The log odds ratio is $log(\frac{p}{1-p})$, where $p$ is the proportion of allocating £20 to the masked trustee.). The results are broadly replicated by linear probability models (See Supplementary Materials S7 File).

Our regressions confirm that participants were significantly more likely to give £20 to the masked counterpart (Model 1: odds ratio = 2.50, 95% CI [1.81, 3.53], $z$ = 5.40, $p$ < .001). To determine whether these decisions were influenced by perceptions of trustworthiness, we incorporated the differences in trustworthiness ratings between masked and unmasked counterparts into the model as a predictor. The greater this difference, the greater the likelihood that the masked trustee received £20 (Model 2: odds ratio = 1.11, 95% CI [1.06, 1.17], $z$ = 4.06, $p$ < .001). For each one-point increase in the trustworthiness rating difference, the odds of the masked individual receiving £20 rose by 11%.

To test if the preference for giving £20 to the masked counterpart was related to the participants' own mask usage, we included their answers to the current usage question in Model 3. Participants who reported they still wore facemasks regularly in public indoor spaces were on average 2.44 times more likely to offer £20 to the masked counterparts than participants who no longer wore facemasks (Model 3: odds ratio = 2.44, 95% CI [1.48, 4.59], $z$ = 3.16, $p$ = .002). Overall, 92% of participants who still wore facemasks in public areas chose the masked counterpart to receive £20, while only 62% of those who no longer wore masks did. Participants' beliefs about the protective effects of facemasks were also associated with their decisions: those who reported that facemasks protect only or mostly the wearer were much less likely to offer £20 to the masked counterpart (Model 3: odds ratio = 0.38, 95% CI [0.18, 0.80], $z$ = −2.51, $p$ = .012) than were those who said masks equally protect the wearer and others. This is consistent with the idea that mask wearers value mask wearing and consequently put more faith in those who also wear masks. It is also consistent with homophily, with trustors preferring to give the larger amount to people like themselves.

**Trustworthiness ratings.** A 2 (masked vs unmasked counterpart) by 2 (male vs female participant) mixed ANOVA was conducted to examine the effects of masks on trustworthiness ratings. There was a significant main effect of facemask ($F$(169) = 22.85, $p$ < .001). As can be seen in Fig 6, on average, trustworthiness ratings were 6.89 points higher for masked counterparts than for unmasked ones ($t$(169) = −4.78, 95% CI[−9.72 points, −4.04 points], $p$ < .001). This finding was consistent with Experiments 1 and 2, suggesting that people who wear facemasks were considered more trustworthy than people who do not, and this effect was still observable sometime after the lifting of mandatory facemask wearing.

**Table 3. Logistic regression predicting whether the masked counterpart received £20 based on Gender, Mask wearing by participant, and difference in Trustworthiness rating between masked and unmasked counterpart.**

| Predictors | Model 1 | | | Model 2 | | | Model 3 | | |
|---|---|---|---|---|---|---|---|---|---|
| | Odds Ratio | CI | p | Odds Ratios | CI | p | Odds Ratio | CI | p |
| (Intercept) | 2.50 | [1.81, 3.53] | <0.001 | 2.94 | [1.72, 5.65] | <0.001 | 6.19 | [3.35, 12.93] | <0.001 |
| Gender | 1.08 | [0.77, 1.50] | 0.665 | 1.37 | [0.92, 2.07] | 0.121 | 1.07 | [0.74, 1.54] | 0.720 |
| Frame | 1.07 | [0.76, 1.49] | 0.703 | 1.21 | [0.82, 1.82] | 0.337 | 1.19 | [0.82, 1.73] | 0.362 |
| Mask wearing habit | | | | 2.21 | [1.30, 4.21] | 0.007 | 2.44 | [1.48, 4.59] | 0.002 |
| Trustworthiness Difference | | | | 1.11 | [1.06, 1.17] | <0.001 | | | |
| Facemask protection attitude: | | | | | | | | | |
| Only & Mostly others | | | | | | | 3.09 | [0.52, 59.46] | 0.303 |
| Only & Mostly themselves | | | | | | | 0.38 | [0.18, 0.80] | 0.012 |
| Observations | 171 | | | 171 | | | 171 | | |
| $R^2$ Tjur* | 0.003 | | | 0.310 | | | 0.160 | | |
| AIC | 212.3 | | | 159.515 | | | 187.046 | | |

*Tjur's [46] coefficient of determination.

## Discussion

In Experiment 3, most people preferred to allocate £20 to the masked rather than the unmasked counterpart. Further analysis explored the factors influencing participants' decisions. Those who reported regularly wearing facemasks in public spaces were more likely to offer £20 to their masked counterparts, but even those that did not still favoured the masked counterpart.

Crucially, differences in trustworthiness ratings between the masked and unmasked counterpart were significantly related to participants' decisions. Specifically, larger rating differences were associated with a higher likelihood that the masked individual was chosen to receive £20.

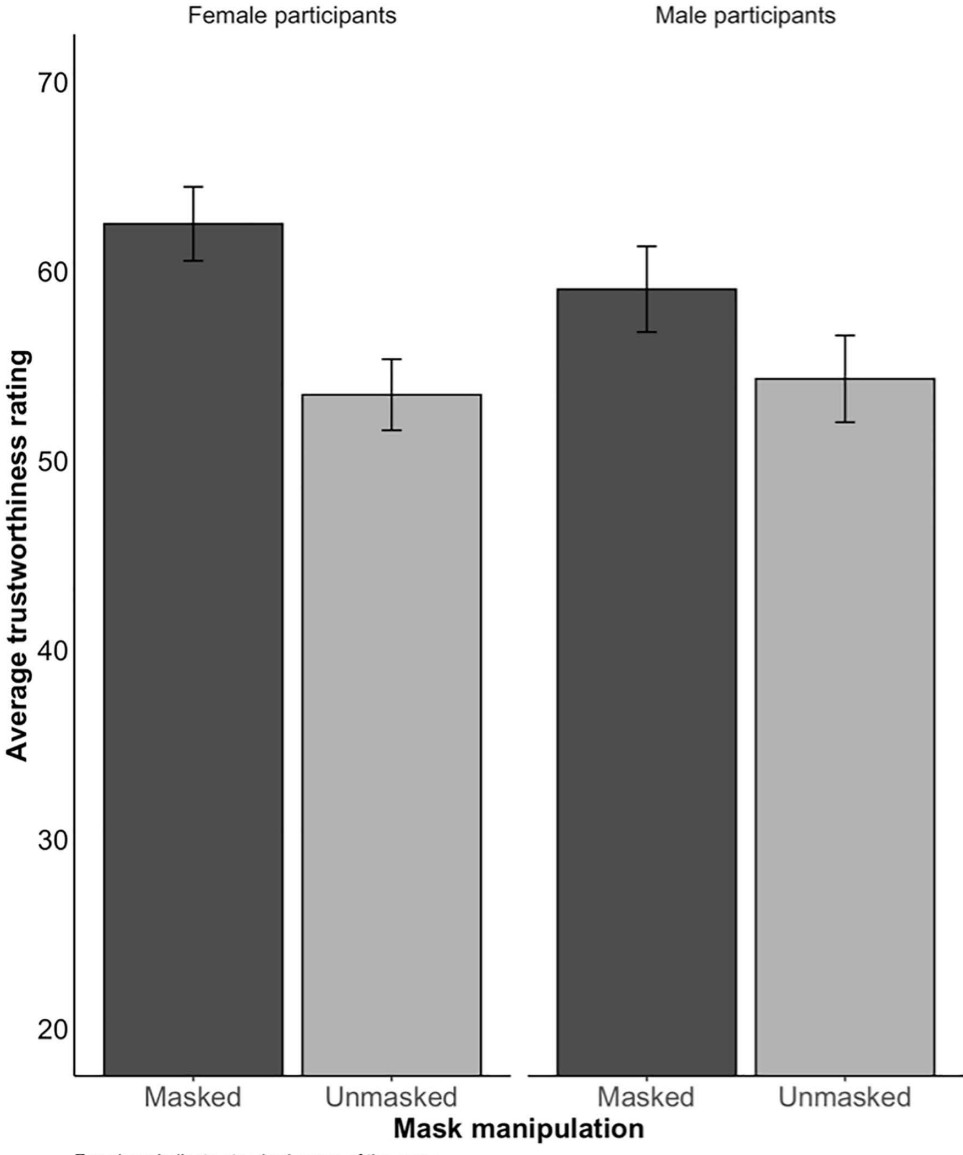

Error bars indicate standard errors of the mean

**Fig 6. Average trustworthiness ratings in Experiment 3, as a function of whether the counterpart was Masked (dark) or Unmasked (light), and whether the participant was Male (right column) or Female (left column).**

## General discussion

In three preregistered experiments, we found that masked faces were consistently rated more trustworthy than unmasked ones. Experiments 1 and 2 looked at stylised financial interactions represented as simple trust games in which the achievement of a beneficial outcome relied on placing trust in a counterpart. This was a between-subjects design in which each participant saw only one counterpart, masked or unmasked. We found that, while masked trustees were perceived as more trustworthy, they were not allocated more money. Masked trustors did not receive more money back from trustees. A plausible reading of this result is that mask wearing had relatively low salience, partly because at the time the experiments were conducted mask wearing was routine. In Experiment 3, when we made the contrast between mask wearing and non-wearing more salient, not only were masked counterparts judged as more trustworthy, but these differential trustworthiness ratings also translated into a decisive preference for the masked counterpart. This tendency was accentuated for participants who still regularly wore masks themselves, but it was seen even in non-mask wearers.

Our finding that mask wearers were judged as more trustworthy than non-mask wearers is in line with studies conducted in various countries during the pandemic (e.g., [23–25]), although they contrast with one study conducted in the US where attitudes towards facemasks were more polarised [47].

The trust-enhancing effect of masks may have been reinforced by the extensive use of social norm messages by governments and public health authorities in promoting the use of facemasks. In the UK, for instance, the National Health Service's slogan was "I wear this to protect you. Please wear yours to protect me." This emphasised social responsibility and contributed to the characterisation of mask wearers as people who did "the right thing", and hence were deserving of trust. Similarly, the French Académie Nationale de Médecine evoked the same idea by means of the Three Musketeers' slogan "All for one, one for all" [48] to reinforce what they called the "altruistic" principle. More broadly, such positive attitudes towards face coverings mark a distinct change from the prior negative perceptions of covered faces recorded in some Western countries (e.g., [19]). It is noteworthy that, under pandemic conditions, mask wearing was linked to increased perceived trustworthiness, aligning with Thielmann and Hilbig's [49] findings that trust decisions are driven more by individual preferences than norms, suggesting that extensive facemask advertising likely shifted public perception and overcame previous negative biases.

Our findings are also in line with the idea that trust and (perceived) trustworthiness are not unitary constructs. Mayer et al. [50] suggested that trust may have an ability dimension, based on the trustor's belief about the trustee's competence in performing trustworthy behaviours, a benevolence dimension, reflecting how much the trustor believes the trustee cares about the trustor's wellbeing, and an integrity dimension, capturing how far the trustor believes the trustee behaves ethically. For reasons discussed already, mask wearing may connect to the integrity dimension, as mask wearers may be seen as possessing positive moral traits. It may connect to the benevolence dimension, to the extent that wearers intend to protect others. When it comes to the ability to engage in trustworthy behaviours, however, situation-specific aspects may be more important. For instance, while we find that the choice between trusting a masked or an unmasked stranger is resolved in favour of the former, our first two experiments demonstrate this increased trust does not inevitably lead to a willingness to risk more money on that stranger's benevolence.

By showing that mask wearing can affect trust and behaviours that depend on it, we add to the growing evidence that practices intended for particular policy purposes may have spillovers on a variety of other behaviours. As observed during the pandemic, particularly in the USA, Europe and Australia, mask wearing can be charged with political meaning and used, perhaps deliberately, as a signal of a variety of traits, such as social identity, political partisanship and in-group status (e.g., [51–56]). An interesting question for future research is whether the enhanced perceived trustworthiness generated by mask wearing can be used, perhaps with deceptive intents, in a deliberate fashion.

## Conclusions

The widespread adoption of facemasks during the pandemic transformed interpersonal perceptions and interactions, illustrating the critical role of public perception in shaping compliance with government emergency mandates [57]. Our

research makes two key contributions. First, we add to the growing evidence that facemasks, at least under pandemic conditions, may serve as a visual signal of trustworthiness and social responsibility. This finding is particularly relevant as mask-wearing has persisted at higher rates in post-pandemic times compared to pre-pandemic levels, as demonstrated in both Italy and several Asian cities, where mask use remains a common practice despite the absence of mandates [58,59].

Second, our research extends the existing literature by examining how this perceived trustworthiness translates into behaviour, specifically in joint and separate evaluation contexts. Under separate evaluation, while masked individuals were perceived as more trustworthy, this perception did not translate into greater monetary trust. However, in joint evaluations—where participants were able to directly compare masked and unmasked counterparts—trustworthiness perceptions significantly influenced behaviour, with more trust placed in the masked individuals. This distinction is critical because, in many everyday contexts—such as at checkouts, salons, or classrooms—people may encounter both masked and unmasked individuals. Our findings suggest that direct comparisons between masked and unmasked individuals can alter trust-related decisions, emphasizing the key role of context in shaping trust dynamics.

## Supporting information

**S1 File. Face stimuli selection.**
(DOCX)

**S2 File. Regression tables of exploratory analysis.**
(DOCX)

**S3 File. Sample size using different exclusion criteria and comprehension questions.**
(DOCX)

**S4 File  Regression results with sample size after applying all four comprehension questions.**
(DOCX)

**S5 File. Analysis of attractiveness ratings.**
(DOCX)

**S6 File. Logistic regression results after re-parameterizing the dependent variable.**
(DOCX)

**S7 File. Linear probability regression results.**
(DOCX)

**S8 File. Regression results restricted to participants who played the trustor first and participants who played the trustee first.**
(DOCX)

**S9 File. Analysis at each transfer level.**
(DOCX)

## Author contributions

**Conceptualization:** Junning Peng, Andrea Isoni, Ashley Luckman, Ivo Vlaev, Dawn Eubanks, Hossam Zeitoun, Daniel Read.

**Data curation:** Junning Peng, Andrea Isoni, Ashley Luckman, Ivo Vlaev, Dawn Eubanks, Hossam Zeitoun, Daniel Read.

**Formal analysis:** Junning Peng, Ashley Luckman.

**Funding acquisition:** Andrea Isoni, Ivo Vlaev, Dawn Eubanks, Hossam Zeitoun, Daniel Read.

**Methodology:** Junning Peng, Andrea Isoni, Ashley Luckman, Ivo Vlaev, Dawn Eubanks, Hossam Zeitoun, Daniel Read.

**Project administration:** Junning Peng, Andrea Isoni, Hossam Zeitoun, Daniel Read.

**Software:** Junning Peng, Andrea Isoni, Ashley Luckman, Daniel Read.

**Supervision:** Andrea Isoni, Daniel Read.

**Visualization:** Junning Peng, Andrea Isoni, Ashley Luckman, Daniel Read.

**Writing – original draft:** Junning Peng, Andrea Isoni, Ashley Luckman, Dawn Eubanks, Hossam Zeitoun, Daniel Read.

**Writing – review & editing:** Junning Peng, Andrea Isoni, Ashley Luckman, Ivo Vlaev, Dawn Eubanks, Hossam Zeitoun, Daniel Read.

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
