## [Decision Letter · Decision Letter 0]

12 May 2025

Dear Dr. Peng,

Thank you for submitting your manuscript to PLOS ONE. After careful consideration, we feel that it has merit but does not fully meet PLOS ONE’s publication criteria as it currently stands. Therefore, we invite you to submit a revised version of the manuscript that addresses the points raised during the review process.

I also reviewed your manuscript throughout. Overall, I also agree with the reviewer's comments. Especially, I hope you address the issue about the data availability policy. Please show clearly the descriptions about data availability, such as a link to a repository, in the main text.

Please check the reviewer's comments and revise your manuscript according to the comments with point-by-point responses.

We look forward to receiving your revised manuscript.

Kind regards,

Yutaka Horita

Academic Editor

PLOS ONE

For additional information about PLOS ONE ethical requirements for human subjects research, please refer to http://journals.plos.org/plosone/s/submission-guidelines#loc-human-subjects-research .

 [This work was supported by the Economic and Social Research Council via the Network for Integrated Behavioural Science [award number: ES/P008976/1].]. 

Reviewers' comments:

Reviewer's Responses to Questions

**Comments to the Author**

1. Is the manuscript technically sound, and do the data support the conclusions?

Reviewer #1: Yes

2. Has the statistical analysis been performed appropriately and rigorously?

Reviewer #1: Yes

3. Have the authors made all data underlying the findings in their manuscript fully available?

Reviewer #1: No

4. Is the manuscript presented in an intelligible fashion and written in standard English?

Reviewer #1: Yes

Reviewer #1: ****************************** ****************************** ******************************

Please find the attached report. *********************************

****************************** ******************************

**Do you want your identity to be public for this peer review?** For information about this choice, including consent withdrawal, please see our Privacy Policy

Reviewer #1: No

---

## [Author Response · Author response to Decision Letter 1]

27 Jul 2025

All comments suggested by the reviewer and editor have been addressed and can be found in the Response to Reviewers and Revised Manuscript with Track Changes.

---

## [Decision Letter · Decision Letter 1]

24 Aug 2025

How Facemasks Shape Trust in Social Interactions

PONE-D-25-08765R1

Dear Dr. Peng,

We’re pleased to inform you that your manuscript has been judged scientifically suitable for publication and will be formally accepted for publication once it meets all outstanding technical requirements.

Kind regards,

Yutaka Horita

Academic Editor

PLOS ONE

Additional Editor Comments (optional):

Reviewers' comments:

Reviewer's Responses to Questions

**Comments to the Author**

Reviewer #1: All comments have been addressed

2. Is the manuscript technically sound, and do the data support the conclusions?

Reviewer #1: Yes

3. Has the statistical analysis been performed appropriately and rigorously?

Reviewer #1: Yes

4. Have the authors made all data underlying the findings in their manuscript fully available?

Reviewer #1: Yes

5. Is the manuscript presented in an intelligible fashion and written in standard English?

Reviewer #1: Yes

Reviewer #1: I thank the authors for addressing my comments in a clear manner. I endorse publication.

Congratulations on a very interesting article.

**Do you want your identity to be public for this peer review?** For information about this choice, including consent withdrawal, please see our Privacy Policy

Reviewer #1: **Yes: ** César Mantilla

---

## [Editor Report · Acceptance letter]

PONE-D-25-08765R1

PLOS ONE

Dear Dr. Peng,

I'm pleased to inform you that your manuscript has been deemed suitable for publication in PLOS ONE. Congratulations! Your manuscript is now being handed over to our production team.

Kind regards,

on behalf of

Dr. Yutaka Horita

Academic Editor

PLOS ONE